# Anesthetic Efficacy of Dexmedetomidine versus Midazolam When Combined with Remifentanil for Percutaneous Transluminal Angioplasty in Patients with Peripheral Artery Disease

**DOI:** 10.3390/jcm8040472

**Published:** 2019-04-06

**Authors:** Ji-Hyoung Park, Sarah Soh, Young-Lan Kwak, Bosung Kim, Sohyun Choi, Jae-Kwang Shim

**Affiliations:** 1Department of Anesthesiology and Pain Medicine, Yonsei University Wonju College of Medicine, Wonju, Gangwon-do 26426, Korea; killerjhjh@yonsei.ac.kr (J.-H.P.); LAGOZZANG@naver.com (B.K.); 2Department of Anesthesiology and Pain Medicine, and Anesthesia and Pain Research Institute, Severance Cardiovascular Hospital, Yonsei University College of Medicine, 50-1 Yonsei-ro, Seodaemun-gu, Seoul 03722, Korea; YEONCHOO@yuhs.ac (S.S.); YLKWAK@yuhs.ac (Y.-L.K.); 3Center of Biostatistics, Wonju College of Medicine, Yonsei University, Wonju, Gangwondo 26426, Korea; CSHH8023@naver.com

**Keywords:** percutaneous transluminal angioplasty, dexmedetomidine, satisfaction

## Abstract

Anesthesia for percutaneous transluminal angioplasty (PTA) involves a high-risk population having a broad spectrum of pain character and intensity. This study delved the anesthetic efficacy of dexmedetomidine versus midazolam, when used with remifentanil. Seventy patients scheduled for femoropopliteal PTA were randomized into two groups receiving either intermittent midazolam boluses (0.03–0.05 mg/kg) (MR group) or dexmedetomidine 0.2–0.7 μg/kg/h after a loading dose of 1.0 μg/kg for 10 min (DR group), both with remifentanil. The primary endpoint was the patients’ satisfaction (1–5, 5; extremely satisfied). Secondary endpoints included postprocedural pain scores (0–10, 10; worst imaginable pain) and adverse events. The satisfaction level of patients was significantly greater in the DR group compared with the MR group (4.0 [3.0, 5.0] versus 4.0 [2.0, 5.0] *p* = 0.021). The number of patients having a postprocedural pain score of at least 3 was significantly greater in the MR group compared with the DR group (10 [29%] versus 2 [6%], *p* = 0.013). The number of patients with hypotensive episodes was higher in the DR group (5 [14.7%] versus 0, *p* = 0.025), which could all be restored with ephedrine. The use of dexmedetomidine in conjunction with remifentanil may be a safe option that provides excellent patient satisfaction while potentially attenuating postprocedural pain.

## 1. Introduction

Confronting an ageing society, the prevalence of peripheral artery disease (PAD) and its related morbidities continue to escalate [1]. Accordingly, the need for a proper anesthetic care for patients undergoing percutaneous transluminal angioplasty (PTA) is increasing [2]. Currently, many cardiovascular procedures that require monitored anesthetic care (MAC) are performed using a remifentanil-based regimen for its superb analgesic efficacy and pharmacokinetic advantage of rapid elimination [3,4]. However, opioids inevitably accompany respiratory depression and do not provide sedation [5,6]. On the other hand, sedatives such as midazolam or propofol have no analgesic effects, and their combined use with opioids further increases the risk of respiratory compromise requiring advanced airway management [7]. In addition, anesthetic care for PTA can be complicated by the following. First, by sharing same risk factors with cardiovascular diseases, patients with PAD exhibit a 2- to 6-times greater risk of cardiovascular events [1,8], comprising a high-risk population. Second, patients with PAD present broad spectrums of pain intensity and character [1], not to mention that endovascular ballooning elicits considerable ischemic pain. Moreover, pain may persist even after 24 h of revascularization, possibly due to oxidative stress and inflammatory response [9]. Thus, validation of an anesthetic regimen that enables maintaining a fine balance between patients’ safety and satisfaction would be of high priority.

Dexmedetomidine is a highly selective alpha 2 agonist that possess unique properties as it is able to provide both sedation and analgesia whilst preserving the respiratory function [10]. Previous studies showed its favorable effects on respiration and interventionists’ satisfaction in catheter ablation for atrial fibrillation [4]. In terms of PTA, dexmedetomidine may also prove to be beneficial for post reperfusion pain as it has been shown to exert anti-oxidant and anti-inflammatory effects in animal models of ischemia-reperfusion injury [11,12], yet, no comprehensive evidence exists in that regard. Thus, we hypothesized that the addition of dexmedetomidine to a remifentanil-based MAC regimen for PTA would improve patients’ satisfaction without respiratory compromise and have the potential of extended analgesic efficacy into the postprocedural period.

The primary aim of this randomized, controlled study was to compare the anesthetic efficacy of dexmedetomidine versus midazolam in PTA by comparing the patients’ satisfaction. Secondary endpoints were interventionists’ satisfaction, pain intensity and postprocedural analgesic requirements up to 24 h after the procedure, and the occurrence of drug-related adverse events.

## 2. Methods

### 2.1. Study Population

The current trial was approved by the institutional review board of the Yonsei University Health System, Seoul, South Korea and then enlisted in http://clinical trials.gov (NCT 02929095) and followed CONsolidated Standards of Reporting Trials (CONSORT) 2010 guidelines. After acquiring informed consent from each patient, 70 patients with American Society of Anesthesiologist physical status I to III, aged 20–80 years, who underwent PTA for femoropopliteal lesions under MAC between October 2016 and March 2018 were enrolled. We excluded patients with psychiatric disorder, cognitive impairment, myocardial infarction and/or stroke, hepatic dysfunction, or congestive heart failure. Patients were randomly and evenly assigned to either midazolam plus remifentanil (MR, *n* = 35) group or dexmedetomidine plus remifentanil (DR, *n* = 35) group by a computerized randomization table. Blinding of the group designation was maintained to the patients and the attending anesthesiologists and interventionists, while the studied drugs were prepared by an anesthesia nurse who was not involved in patient care or assessment.

### 2.2. Anesthetic and Procedural Management

After entering the hybrid operating theatre, ECG, non-invasive blood pressure, pulse-oximetry (SaO_2_), and respiratory rate were monitored in all patients. Before starting MAC, all patients received oxygen at 5 L/min through an oxygen mask. Then, MAC was initiated using either MR or DR regimen and procedural draping and local anesthesia with 1% lidocaine to the ipsilateral and/or contralateral inguinal area were followed after 10 min of commencing MAC.

MAC was provided to all patients as follows. In the MR group, 0.03–0.05 mg/kg of midazolam was given as a bolus placement, while the patients in the DR group received same amount of 0.9% saline. In the DR group, 1.0 μg/kg of dexmedetomidine bolus placement was done for 10 min using a syringe pump, and then maintained until the end of the procedure at infusion rates of 0.2–0.7 μg/kg/h, while the MR group received same infusions of 0.9% saline. Dose adjustments were done to target the Ramsay sedation score of 2 to 4 (1 = anxious and agitated, restless or both; 2 = co-operative, oriented and tranquil; 3 = response to commands only; 4 = brisk response and 5 = sluggish response to light glabellar tap or loud auditory stimulus; 6 = no response to stimulation) [13].

In all patients, continuous infusion of remifentanil at 1.2 μg/kg/h remifentanil using a syringe pump was started concomitantly with the bolus placement of midazolam or dexmedetomidine. The infusion rate of remifentanil was adjusted according to the pain score evaluation (11-point numeric rating scale; 0 = no pain to 10 = worst imaginable pain). The remifentanil infusion rate was increased by 0.6 μg/kg/h and up to 7.2 μg/kg/h to maintain the pain score ≤3. Remifentanil infusion rate was decreased by 0.6 μg/kg/h until reaching 0.6 μg/kg/h when the pain score was between 0 and 1.

If hypotension occurred (mean arterial pressure < 60 mmHg), 4–8 mg of ephedrine was given intravenously. If bradycardia (heart rate < 45 bpm) occurred, 0.1–0.2 mg of glycopyrrolate was given intravenously. If bradycardia persisted, 0.5 mg of atropine was given intravenously. If respiratory depression (respiratory rate < 10/min) and/or hypoxia (SaO_2_ < 90%) occurred, patients were given a verbal awakening stimulation (calling their name) followed by a gentle squeeze at the trapezius muscle if unresponsive. If respiratory depression persisted despite these stimulations, the rate of remifentanil infusion was reduced by 0.6 μg/kg/h and head extension with or without jaw thrust was performed, while preparing for further advanced airway management.

PTA was considered successful when the degree of residual stenosis was below 30% without any dissection compromising the flow.

In the postprocedural period, patients were given acetaminophen, tramadol, meperidine, or oxycodone as rescue analgesics upon their request or when the pain score exceeded 4. The choice of the rescue analgesic was done at the discretion of the attending physician at the ward. In order to compare the doses, substitution was made with equipotent doses of morphine.

### 2.3. Primary Endpoint and Assessment

The primary endpoint of the current study was to compare the anesthetic efficacy between DR and MR in terms of patients’ satisfaction about the MAC. At the post-anesthesia care unit (PACU), the patients were asked to choose their satisfaction score about their given MAC using a 5-point numerical rating scale (1 = extremely dissatisfied, 2 = dissatisfied, 3 = neutral, 4 = satisfied, and 5 = extremely satisfied) before being transferred to the general ward.

### 2.4. Secondary Endpoints and Assessments

After the end of procedure, the interventionists were asked to choose their satisfaction score using the same 5-point numeric rating scale. In the PACU before being transferred to the general ward, patients were asked for the overall pain score during the procedure using the above described 11-point numeric rating scale. They were also asked for the overall pain score during the 24 h of the post-procedural period. Patients were also asked for the frequency of having a pain score ≥3 during the study period. The number of patients requiring rescue analgesics, and the total amount of analgesics administered to the patients in morphine equivalent dose was recorded.

### 2.5. Statistical Analysis

Sample size calculation was calculated based on the patients’ satisfaction score. In a previous study, the level of satisfaction by MR regimen in catheter ablation for atrial fibrillation was 2.9 ± 0.6 [4]. Assuming that the use of dexmedetomidine instead of midazolam can enhance the satisfaction level by 0.5 or more, the estimated number of patients in each group was 31 (α = 0.05 and power = 0.9). Accounting for a dropout rate of 10%, we decided to enroll 35 patients in each group.

All statistical analyses were performed with the SAS 9.4 and Rx64 program. Intergroup comparisons of satisfaction and pain scores were done by Mann-Whitney U test. Continuous variables were assessed for their distribution (Shapiro-Wilk test). Intergroup comparison of other variables that showed normal distribution were tested using the independent *t*-test (mean ± standard deviation [SD]). Skewed data were tested using Mann-Whitney U test (median [interquartile range]). Intergroup comparisons of categorical variables were tested using Chi-square test (*n* [%]). For pain scores that were assessed at 3 time points, post hoc Bonferroni correction was applied. Thus, the *p* values for the pain scores were considered statistically significant when <0.017. Otherwise, *p* < 0.05 was considered statistically significant.

## 3. Results

A total of 105 patients were screened, and 70 among them were enrolled and evenly randomized into either the MR or DR group. There was one dropout among the 70 enrolled subjects (Figure 1).

Patients’ characteristics and procedural data are displayed in Table 1. Patients’ satisfaction (5-point numeric rating scale, 5 = extremely satisfied), which was the primary endpoint, was significantly greater in the DR group compared with the MR group (4.0 [3.0,5.0] versus 4.0 [2.0,5.0] *p* = 0.021) (Figure 2, Table 2). The satisfaction of the interventionists did not differ between the groups (Table 2). When analyzing patients with considerable pain at rest (Rutherford category ≥4 having at least ischemic pain at rest), the patients’ satisfaction score was more evidently in favor of the DR group than the MR group (4.0 [4.0,5.0] versus 3.5 [3.0,4.0], *p* = 0.046).

Hemodynamic data including heart rate and mean arterial pressure, and SaO_2_ were all within clinically acceptable ranges and showed no intergroup differences throughout the procedure (Figure 3).

In terms of drug-related adverse events, incidences of bradycardia and hypoxia were not different between the groups, while the number of patients with 1 or more hypotensive episodes was higher in the DR group compared with the MR group, which could all be managed by a single bolus of intravenous ephedrine administration (Table 2).

Pain scores and the frequency of rescue analgesic requirement up to 24 h after the procedure were similar between the groups. However, the number of patients having a pain score of at least 3 was significantly greater in the MR group compared with the DR group (10 [29%] versus 2 [6%], *p* = 0.013), while the total amount of administered rescue analgesics in morphine equivalent dose showed a trend towards being lower in the DR group compared with the MR group (6.6 [5.0, 10.0] mg versus 10.0 mg [5.5, 18.2] mg, *p* = 0.062) (Table 2).

## 4. Discussion

Anesthetic care for PTA can be challenging as patients with PAD comprise a high-risk group for cardiovascular events [14,15]. Thus, emphasis should be given to prevent anxiety or pain responses that cause maladaptive sympathetic activation during PTA. Also, successful PTA requires immobilization while confronting the fact that the patients already exhibit a broad spectrum of pain in addition to the ischemic pain elicited by intermittent ballooning.

For MAC during cardiovascular procedures, continuous infusion of remifentanil has become the mainstay of anesthesia with the intermittent use of sedatives as necessary. Opioids have their advantages in that they are potent analgesics able to cover diverse pain characters and they lack direct myocardial depressant effects except for their central vagotonic influence [16]. However, in patients with rigorous pain, increasing the dose of remifentanil may not be feasible as opioids cause respiratory depression, which could be troublesome in a supine position with an increased risk of airway-compromise [5]. On the other hand, increasing the dose of sedatives may not suffice as the commonly used sedatives lack analgesic effects, and possibly also hyperalgesia lowering the pain thresholds [17], on top of increasing the risk of respiratory depression. Thus, finding an anesthetic regimen that provides adequate sedation without suppressing respiration would be of high priority for PTA.

Dexmedetomidine is a potent and selective alpha 2 agonist exerting sedative and analgesic effects at the same time without compromising respiration. Accordingly, emerging evidence supports its beneficial influence in critical care and MAC involving conscious sedation for cardiovascular procedures [4], while its anesthetic efficacy in PTA has never been validated heretofore. Moreover, nociceptive pain with PAD involves intermittent episodes of ischemia-reperfusion and its chronic manifestation has been shown to result in complex neuropathic pain as well [18]. Importantly, even successful PTA has been shown to elicit significant postprocedural pain related to reperfusion and oxidative stress [9,19]. Experimentally, dexmedetomidine has been shown to be effective in neuropathic pain through the inhibition of IL-6 and TNF-a [20]. Also, dexmedetomidine has been shown to attenuate oxidative and inflammatory stress responses related to ischemia-reperfusion [11,12]. Moreover, the EEG patterns observed during dexmedetomidine infusion more closely resembles those of the natural sleep as opposed to other anesthetics [21], which may be favorable in terms of the sleep quality. Thus, we hypothesized that the use of dexmedetomidine in PTA would not only improve patients’ satisfaction during the procedure, but also exert beneficial influence in postprocedural pain.

As our results indicate, patients receiving dexmedetomidine were more satisfied with their anesthetic care than those receiving midazolam with none of the patients in the DR group reporting a satisfaction score of less than 3 (from a 5-point numeric scale with 5 being extremely satisfied) confirming our primary hypothesis. This beneficial effect of dexmedetomidine was even more evident in patients with considerable resting pain (Rutherford category ≥4). In terms of its efficacy on pain scores, we could only observe trends towards lower procedural and postprocedural pain scores in the DR group compared with the MR group. Interestingly, there were a significantly lower number of patients in the DR group who experienced a significant postprocedural pain (pain scores of ≥3) than the MR group, despite receiving a lower amount of rescue analgesics. Thus, per our hypothesis, the experimentally proven anti-inflammatory property of dexmedetomidine attenuating oxidative stress may have been responsible for the reduced incidence of intense pain after reperfusion in the current study. Also, concerns have been raised regarding a potential hyperalgesic phenomenon after discontinuation of remifentanil infusion that is not clearly understood [22]. Dexmedetomidine may also attenuate the hyperalgesic response after the discontinuation of remifentanil infusion by modulating spinal cord *N*-methyl-d-aspartate receptor activation via suppression of NR2B subunit phosphorylation [23], which may have contributed to the attenuated postprocedural pain.

In terms of drug-related adverse events, use of dexmedetomidine was associated with more frequent episodes of hypotension, which could all be rapidly restored by a single ephedrine bolus. This result agrees with our previous report involving patients undergoing catheter ablation of atrial fibrillation [4]. By being a sympatholytic, dexmedetomidine’s potential to cause hypotension and bradycardia has been well acknowledged. Yet, its biphasic hemodynamic response usually shows an initial increase by mean arterial pressure with subsequent bradycardia and return to baseline hemodynamic after stabilization [24]. As with our previous study [4], the chosen dose range of our study seems to be safe in terms of mean arterial pressure and heart rate, as they were similar between the studied drugs, and showed that the hypotensive events could all be rapidly corrected.

In terms of opioid sparing effect and respiratory function, we could not observe any significant favorable influence of dexmedetomidine over the conventional use of midazolam, which may be attributable to the following. The chosen MR regimen has been used and adjusted in our institution over the past 5 years for PTA to provide safe conscious sedation, which yielded acceptable patient satisfaction as well.

The limitations of the current study are as follows. Although there are solid experimental backgrounds, we did not measure markers of oxidative stress or inflammation, and thus, we can only speculate about the favorable influence of dexmedetomidine on post-reperfusion pain. Also, the sample size may be insufficient for validating our secondary endpoints, especially in terms of pain scores as they showed only statistical trends towards being lower in the DR group. Lastly, due to the nature of the disease, the inclusion of male patients was predominant, limiting the extrapolation of these results to female patients.

In conclusion, MAC for a high-risk group requires special attention balancing the safety and the patients’ satisfaction. Also, a proper anesthetic regimen should be tailored to cover the disease- and procedure-specific pain characteristics that are unique to patients with PAD undergoing PTA. In line with these needs, the current study provides primary evidence that the use of dexmedetomidine in conjunction with remifentanil may be a safe option that provides excellent patients’ satisfaction while potentially attenuating postprocedural pain as well.

## Figures and Tables

**Figure 1 jcm-08-00472-f001:**
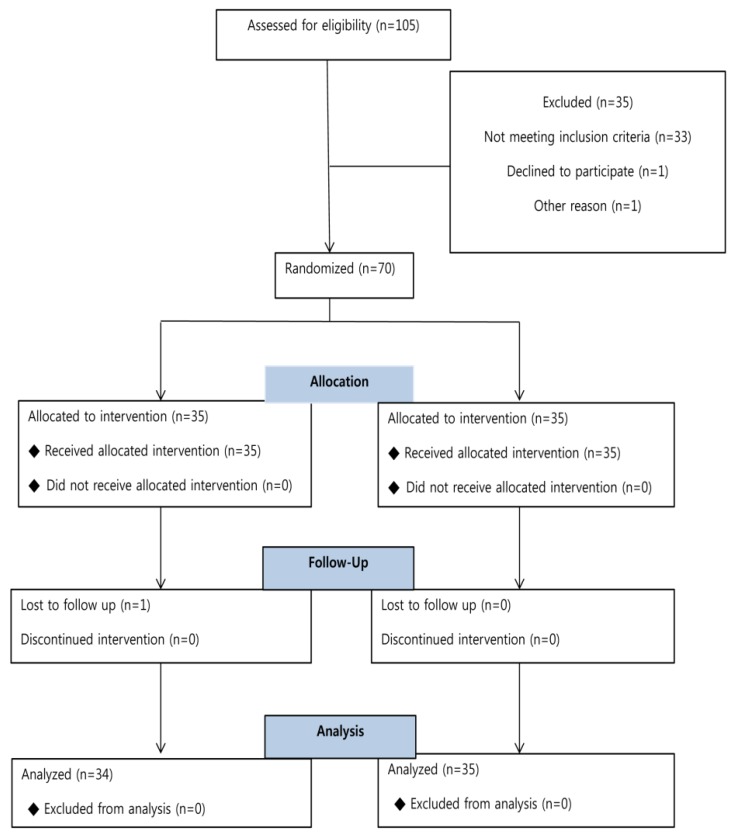
Research flow chart.

**Figure 2 jcm-08-00472-f002:**
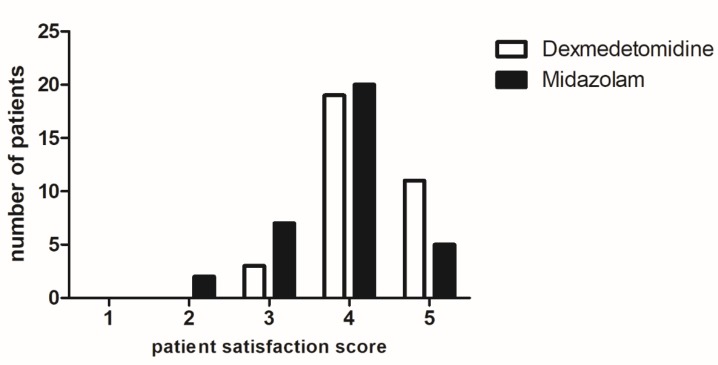
Patient satisfaction score.

**Figure 3 jcm-08-00472-f003:**
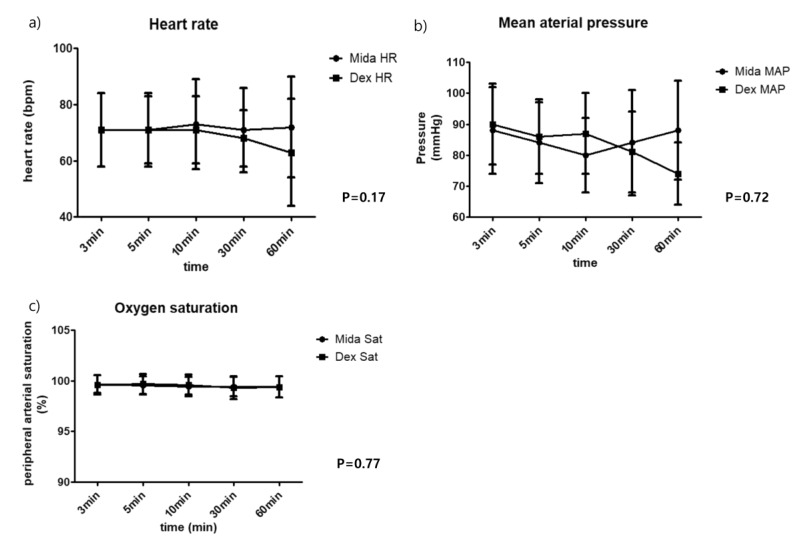
Hemodynamic data: (**a**) heart rate; (**b**) mean arterial pressure; (**c**) oxygen saturation.

**Table 1 jcm-08-00472-t001:** Patients’ Characteristics and procedural data.

	Midazolam (*n* = 35)	Dexmedetomidine (*n* = 34)
Age (years)	64.7 ± 12.5	64.1 ± 11.5
Sex (M/F)	34/1	27/7
Body surface area (m^2^)	1.75 ± 0.16	1.71 ± 1.9
Hypertension	25 (71.4)	20 (58.8)
Diabetes mellitus	19 (54.3)	16 (47.1)
Coronary artery disease	13 (37.1)	11 (32.4)
End-stage renal disease	1 (2.9)	2 (5.9)
Cerebrovascular accident	3 (8.6)	5 (14.7)
Medications		
Aspirin	30 (85.7)	23 (67.6)
Thienopyridine	26 (74.3)	21 (61.8)
Cilostazol	8 (22.9)	3 (8.8)
Warfarin	3 (8.6)	1 (2.9)
Statin	20 (57.1)	19 (55.9)
Beta-blocker	3 (8.6)	3 (8.8)
Rutherford category		
1	1 (2.9)	1 (2.9)
2	4 (11.4)	5 (14.7)
3	18 (51.4)	17 (50.0)
4	5 (14.3)	8 (23.5)
5	4 (11.4)	1 (2.9)
6	3 (8.6)	2 (5.9)
Current smoker, *n* (%)	8 (22.9)	6 (20.0)
Procedure time (min)	59.7 ± 37.8	53.7 ± 31.4
Intraoperative medication		
Remifentanil (µg/kg/h)	0.72 [0.50, 1.31]	0.72 [0.49, 1.06]
Midazolam (µg/kg/h)	20.51 [13.79, 44.44]	0 [0, 0]
Dexmedetomidine (µg/kg/h)	0 [0, 0]	51.00 [37.87, 74.40]

Data are displayed in mean ± SD, or *n* (%).

**Table 2 jcm-08-00472-t002:** Primary and secondary endpoints.

	Midazolam (*n* = 35)	Dexmedetomidine (*n* = 34)	*p*-Value
**Primary endpoint**			
Patients’ satisfaction	4.0 [3.0,4.0]	4.0 [4.0,5.0]	0.021 *
**Secondary endpoints**			
Interventionists’ satisfaction	4.0 [4.0,4.5]	4.0 [3.5,4.0]	0.860
Pain scores			
Baseline	2 [1,3]	2 [2,4]	0.734
Procedural	3 [0,4]	0.5 [0,4]	0.192
Post-procedural 24 h	0 [0,3]	0 [0,1]	0.213
Pain score ≥3	10 (28.6)	2 (5.9)	0.013 *
Rescue analgesics	13 (37.1)	13 (38.2)	0.925
Rescue morphine dose (mg)	10.0 [5.5,18.2]	6.6 [5.0,10.0]	0.062
Adverse events			
Bradycardia	0	0	1.000
Hypotension	0	5 (14.7%)	0.025 *
Hypoxia	1 (2.9%)	0	1.000
Nausea	0	1 (2.9%)	0.310

Data are displayed in mean ± SD, *n* (%), or median (interquartile range). Satisfaction score was assessed using a 5-point numerical scale (1 = extremely dissatisfied~5 = extremely satisfied). * = *p* < 0.05. Pain score was assessed using an 11-point numeric scale (0 = no pain~10 = worst imaginable pain). Rescue analgesic dose was calculated as morphine equivalent dose. * = *p* < 0.05.

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
