# Peer review of "Anesthetic Efficacy of Dexmedetomidine versus Midazolam When Combined with Remifentanil for Percutaneous Transluminal Angioplasty in Patients with Peripheral Artery Disease"

_jcm, 2019, doi:10.3390/jcm8040472_

Reviewer 1 Report

Thank you for requesting my contribution to review the manuscript "Anesthetic Efficacy of Dexmedetomidine versus Midazolam When Combined with Remifentanil for Percutaneous Transluminal Angioplasty in Patients with Peripheral Artery Disease".

The article is well written, and easy to read. The results are not confusing.

Minor revisions could be suggested:

Thank you for presenting the figures and tables at the end of the manuscript

Figure 2: The two groups (DR and MR) should be in the same diagram

Table 1: I suggest removing the p value for comparing groups at this stage. The study is randomized and it is not correct and relevant to do so. If an observed difference is major, an absolute standardized difference (ASD) calculation should be performed.

Incidence of hypotension : Please specify in the results section whether this is the incidence of hypotension episodes or the incidence of patients with 1 or more episodes of hypotension.

Table 2: The comparison of Remifentanil (μg / kg / h), Midazolam (μg / kg / h), Dexmedetomidine (μg / kg / h) and Procedure time (min) should appear in Table 1 (without comparing them with a statistical test) and the rest of Table 2 could be merged with Table 3, with clear distinguishing the primary outcome from the secondary outcomes.

I hope this will help.

Thank you.

Author Response

Answers to Specific Criticisms of the Reviewer 1

We are grateful to the reviewer for the insightful and constructive comments. The answers to your comments are listed below.

 Comments to the Author

The article is well written, and easy to read. The results are not confusing.

Minor revisions could be suggested:

Thank you for presenting the figures and tables at the end of the manuscript

 Figure 2: The two groups (DR and MR) should be in the same diagram

Answer: We have revised the figure as you commented.

 Table 1: I suggest removing the p value for comparing groups at this stage. The study is randomized and it is not correct and relevant to do so. If an observed difference is major, an absolute standardized difference (ASD) calculation should be performed. Table 2: The comparison of Remifentanil (μg / kg / h), Midazolam (μg / kg / h), Dexmedetomidine (μg / kg / h) and Procedure time (min) should appear in Table 1 (without comparing them with a statistical test) and the rest of Table 2 could be merged with Table 3, with clear distinguishing the primary outcome from the secondary outcomes.

Answer: We have revised the Tables and the Results section as you commented.

 Incidence of hypotension: Please specify in the results section whether this is the incidence of hypotension episodes or the incidence of patients with 1 or more episodes of hypotension.

Answer: It was the number of patients with 1 or more episodes of hypotension. We have revised the Abstract and Results section to clarify this matter as you commented.

 Thank you.

Reviewer 2 Report

1) The authors are commended on a novel study evaluating the effects of dexmedetomidine plus remifentanil vs. midazolam plus remifentanil on patient satisfaction and post-procedural pain in those with PAD undergoing PTA. 

2) Please provide the hypothesis in the Intro section.

3) The authors need to justify why appropriate transformations of data (e.g., logarithmic) were not completed when distributions of data were non-normal. Use of non-parametric tests are not ideal.

3) Insufficient number of female patients in the trial makes the results of this study more applicable to males only. This should be included as a limitation in the Discussion section.

4) Minor misalignments in tables.

5) The paper had a number of grammatical errors and typos. A thorough copy editing is needed.

6) The Discussion is too long and difficult to follow. Please reduce by 25-50%.

Author Response

Answers to Specific Criticisms of the Reviewer 2

We are grateful to the reviewer for the insightful and constructive comments. The answers to your comments are listed below.

 Comments to the Author

The authors are commended on a novel study evaluating the effects of dexmedetomidine plus remifentanil vs. midazolam plus remifentanil on patient satisfaction and post-procedural pain in those with PAD undergoing PTA. 

 1) Please provide the hypothesis in the Intro section.

Answer: We have inserted a sentence describing our hypothesis in the Introduction section as you commented. The added sentence is as follows; ‘Thus, we hypothesized that the addition of dexmedetomidine to a remifentanil-based MAC regimen for PTA would improve patients’ satisfaction without respiratory compromise and have the potential of extended analgesic efficacy into the post-procedural period.’

 2) The authors need to justify why appropriate transformations of data (e.g., logarithmic) were not completed when distributions of data were non-normal. Use of non-parametric tests are not ideal.

Answer: We agree with your concern that log-normal transformation of skewed data is one of the most popular methods to decrease the variability of the data. However, this method also has its inherent limitations, and the use of non-parametric tests to deal skewed data is also common for their distribution-free nature, while having a reduced statistical power compared to the parametric tests. The satisfaction score of the patients (our primary endpoint) was compared with Mann-Whitney U test a priori as with the pain scores regardless of their data distribution as these numeric upscale scores are usually tested through this way (we apologize not to have mentioned it in our first draft; we had only mentioned about the pain scores in our first draft). Thus, acknowledging the reduced statistical power to apply non-parametric tests for skewed data, we have used this method as it would not deal our primary endpoint (non-parametric test for skewed data were applied only for the amounts of remifentanil, midazolam, dexmedetomidine, and rescue morphine dose). We have revised the statistical analysis section accordingly to clarify this matter.

 3) Insufficient number of female patients in the trial makes the results of this study more applicable to males only. This should be included as a limitation in the Discussion section.

Answer: We have added this as an additional limitation of the study as you commented. The added sentence is as follows; ‘Lastly, due to the nature of the disease, the inclusion of male patients was predominant that limits the extrapolation of these results to female patients.’

4) Minor misalignments in tables.

Answer: We have revised the Tables accordingly.

 5) The paper had a number of grammatical errors and typos. A thorough copy editing is needed.

Answer: We apologize for our errors. We have rechecked our manuscript by a native English-speaking person and corrected the grammatical and typographical errors as you commented.

 6) The Discussion is too long and difficult to follow. Please reduce by 25-50%.

Answer: We have reduced the Discussion by approximately 30% as you commented

 Thank you.